# Imagining the Post-COVID-19 Polity: Narratives of Possible Futures

**James White McAuley [1],\* and Paul W. Nesbitt-Larking [2]**

1 Department of Behavioural and Social Sciences, School of Human and Health Sciences, University of Huddersfield, Huddersfield HD1 3DH, UK
2 Department of Political Science, Huron University College, London, ON N6G 1H3, Canada
\* Correspondence: j.w.mcauley@hud.ac.uk

**Abstract:** The COVID-19 crisis is arguably the most important development of the 21st century so far and takes its place alongside the great eruptions of the past century. As with any crisis, the current pandemic has stimulated visions and proposals for post-COVID-19 societies. Our focus is on narratives—both predictive and prescriptive—that envisage post-COVID-19 political societies. Combining narrative analysis with thematic analysis, we argue that societal changes conditioned by the pandemic have accelerated a turn toward five inter-related developments: A renaissance in rationality and evidence-based science; a return to social equality and equity, including wage equity and guaranteed incomes; a reimagining of the interventionist state in response to crises in the economy, society, the welfare state, and social order; a reorientation to the local and communitarian, with reference in particular to solidaristic mutual aid, community animation, local sourcing, and craft production; and the reinvention of democracy through deep participation and deliberative dialogical decision making. The empirical focus of our work is an analysis of predominantly legacy media content from the *Canadian Periodicals Index* related to life after the pandemic and post-COVID-19 society.

**Keywords:** COVID-19; pandemic; media; narrative analysis; thematic analysis

## 1. Introduction

Alongside climate change, 9/11, the War on Terror, and the invasion of Ukraine, the pandemic that began in 2020 is a major defining event of the twenty-first century (Kinnvall and Singh 2022; Mahendran et al. 2022). Its impact equals, and perhaps will exceed, the seismic eruptions of the twentieth century: the Great Depression and the two World Wars. The world we thought we knew and the global order, which has been the basis of our collective and structured existences, have evaporated, and the direction of change and ultimate outcome remain to be determined.

It is challenging to find words to express the magnitude of what has happened. There is, of course, grave and abiding danger, which is often hidden or implicit, but there is also opportunity, renewal and revitalisation, and the sense of possibility that emerges in the context of the sudden necessity of doing things dramatically differently. From its earliest stages, the global pandemic put into place profound and thoroughgoing questions surrounding the existing social structure and the impact on polities (political societies). The global pandemic is almost entirely universal. It affects people throughout the world. It is deeply experienced in that most people have had their lives turned upside down through its impact—and more will in the future. It is a thoroughgoing undermining of routines and daily social and personal lives. Furthermore, it has had profound implications for material and economic life.

A vivid illustration of the disruptive and unprecedented characteristics of the pandemic may be found in the Canadian experience and, most recently, the events surrounding the Canadian convoy protests of the Winter of 2022 (Nasrallah 2022; Seto 2022). Mobilised,

funded, and organised by a loose coalition of anti-vaccine, alt-right, Christian right, white nationalist, and other anti-government elements, the convoy was ostensibly a protest against COVID-19 vaccine mandates or restrictions, notably those at the Canada/US border. Sharing characteristics with Capelos et al.'s (2022) concept of *ressentiment*, the protesters demonstrated "moral victimhood, indignation, a sense of destiny and powerlessness" (Capelos et al. 2022). Through coordinated action, it evolved into a series of blockades and occupations, including a large encampment of trucks and temporary structures immediately outside the federal government buildings in Ottawa.

Disavowed by organised labour as well as the Canadian Trucking Alliance, the occupiers made use of industrial sirens and horns, as well as a forthright and occasionally menacing co-presence, to claim public space and attention, while also injecting a carnival atmosphere into the events as an attempt to claim legitimacy as a peaceful protest. Informed by media exposure to right-wing media outlets, such as Fox News, as well as a range of social media conspiracy sites, the predominant orientation toward the legacy or mainstream media (that is, the mass media of established and corporate print-based newspapers, broadcast television, and radio) was one of deep distrust and hostility, resulting in the physical harassment and intimidation of working journalists (Fenlon 2022). The general trend of opinion polls conducted demonstrated that approximately two-thirds of Canadians opposed the objectives and tactics of the occupiers, while about one-third supported them (Abacus Data 2022; Angus Reid 2022).

Our focus in this article is not on the Canadian convoy protests, whose presence here is simply to illustrate through sharp contrast the generally liberal and pluralistic orientations of the mainstream media sources that are the basis of our investigations. These progressive narratives stand in sharp contrast to the reactionary counter-narratives of the Canadian convoy protests. We concentrate on the writings of the mainstream or legacy journalists, along with the invited opinion leaders, who reflected upon and expressed their proposals and visions for the post-COVID-19 polity. For the Ottawa occupiers, these constitute the enemy, the much-derided mainstream media. Very few of the professional journalists or commentators that we referenced supported the reactionary, ultra-libertarian, seditious, or insurgent demands of the occupiers. Instead, their views of the post-COVID-19 polity were either broadly progressive or, as might be anticipated, mainstream and neutral, in the sense of conforming to existing metanarratives of liberal capitalist political society. In brief, while the occupiers looked backward in their counter-narrative frameworks, with a "make Canada great again" theme, the orientation of most journalists and commentators in our study was forward looking, imagining possibility, and assessing probability. In fact, of the 228 sources that were the basis of our study (see methodological details below), 92 of them (40.4 percent) were broadly progressive, 112 were neutral in tone (49.1 percent), 19 (8.3 percent) were unclassifiable, and only 5 (2.2 percent) could be described as reactionary. In other words, there is an almost neutral or progressive character to Canadian journalism and commentary when it comes to envisaging the post-COVID-19 polity.

The principal research aim of our study is exploratory. Through an analysis of relevant themes, in which we blend our own research expectations with an analysis of the data, we aim to portray the ways in which the Canadian legacy media have constructed narratives of the future in conceptualising the post-COVID-19 polity. As social scientists, we do not predict outcomes, particularly when the conditions of the present are so complex, novel, and multifaceted. What we aim to do is to identify evidence of existing trends and tendencies in the current period of transition and upheaval and extrapolate from them. We agree with Sools (2020) that in analysing how the future is imagined, we better understand what motivates and guides ideas and agency in the present. We do not know, and currently cannot know, the outcome of the many deep and complex forces at play in global economies, cultures, and polities, all aimed at marshalling and regulating the immediate necessities of life in western societies under the threat of the global pandemic.

## 2. Theoretical Perspectives

In their reflections on the pandemic and the need for political change, Chomsky and Polychroniou (2021) suggest that neoliberalism, the essential form of capitalist logic that has dominated in the West since the early 1970s, has left contemporary global society ill-prepared for any increase in demands now being made on welfare and wellbeing. The entire system of production, distribution, and exchange has undergone massive strain and stress, which the system experiences through the labour process, labour markets, and the distribution of resources. We would add that the stresses have been cultural and political as well as economic, and that polities are beginning to witness the impact of transformative change. As with all transformations on this scale involving the fate of broadly democratic polities, one can detect both dangers and opportunities. It is possible to see in the current circumstances the seeds of tyranny, authoritarianism, and despotism, on the one hand, or a revitalised pluralism and a more equitable democratic system, on the other hand. Our analysis of Canadian narratives may be situated in a broader global analysis of progressive and reactionary visions of the polity in the COVID-19 era, in particular the European-based research on affective politics and the impact of ressentiment (a complex emotional construct of aggression, grievance, bitterness, victimhood, shame, envy, and frustration) as a reactionary response toward the progressive values of the New Social Movements, notably social equity, feminism, multiculturalism, and diversity (Capelos et al. 2021, 2022; Capelos and Demertzis 2022; Verbalyte et al. 2022).

So, are we at some turning point in history? Is there potential for the rise of new ways of thinking about social structure, perhaps even the growth of social movements focusing on more egalitarian visions of the future? (Mahendran et al. 2022; Squire 2022). Will we simply return to the pre-pandemic state (Carr 2020) brought about by the pre-existing social forces and dominant ideologies? Or do we see the seeds of a turn to reactionary politics, a politics of resentment and ressentiment? Grounded in our own past research (Nesbitt-Larking 2007), our experience of living attentively through the pandemic, and reading the limited scholarship on expectations of or aspirations toward changes in the post-COVID-19 polity since the onset of the pandemic, we believe that the societal changes conditioned by the global pandemic have accelerated a societal turn toward five inter-related developments: (1) rationality and science, (2) social equality, (3) the interventionist state, (4) the local and communitarian, and (5) the deeply participative.

### 2.1. Rationality and Science

The pandemic has rapidly and profoundly revealed the critical importance of evidence-based and scientifically grounded information as well as the need for rationality and systematic order in the provision of public health and other services (Carnegie Civic Research Network 2021; Pleyers 2020; Teovanovic et al. 2021; Zinn 2021a, 2021b).

In the contemporary world of post-ideological and post-structural openness, in which social media have increasingly become forums of toxic disinformation and post-truth rationalisations, populist leaders and movements, such as the Canadian convoy protests, have been able to dominate through bluff, bullying, and deception (Baron 2018; Bruckman 2022; Kinnvall and Singh 2022; Seargeant 2022). They simply make up plausible-sounding responses to any situation, stigmatise and marginalise enemies, deflect attention, distract with absurdities, downplay realities, and denigrate the opposition. For a time, regardless of how ill-informed and incapable the populist leader or movement is, these techniques are functional. Those raising criticism are accused of self-interested bias, drawing upon "fake news", failing to express the correct appreciation, or simply exhibiting bad manners for daring to question the existing "reality". However, the global pandemic cannot be dismissed or explained away with bluff and bully tactics, and populist leaders cannot claim to be experts in it for long. Their fraudulent posturing and inadequate floundering soon become clear to all those who are able to see, even perhaps reluctantly and fitfully, some of their core supporters.

In terms of the polity, this speaks to the resurgence of science and rationality, of the systematic use of evidence and reason in the conduct of human affairs, from pure science to applied science, to epidemiology and public health administration. In such circumstances, public health regimes, pandemic preparedness, universal medical provision, rational systems designed to enhance aggregate public wellness and reduce sickness, as well as a clean environment, become matters of enlightened self-interest for all, including those who currently stand to lose status, privilege, and economic advantage through the changes that are needed to introduce such developments. Given the importance of rationality and science, we anticipate their presence in the post-COVID-19 narratives we investigate.

### 2.2. Social Equality

In certain respects, the spread of highly contagious viruses in a series of interconnected hotspots has necessarily flatlined the social structure and social order. No matter one's status, class, wealth, ethnic identity, or caste, the human body is equally vulnerable to infection and transmission, giving rise to the populist name of the "great equaliser". However, as Ann Phoenix (2022) points out in her contribution to this special issue, no crisis of this kind is ever experienced equally across demographic groups. To begin with, ethno-racial minorities, the poor, women, and the aged are predisposed through their existing life circumstances to be at greater risk of further ill health. Added to this, frontline workers—those at greatest risk—are disproportionately working class, racialised, women, and the marginalised (Golden and Muggah 2020). As the pandemic has grown and inoculations have become available, the capacity to protect, isolate, and quarantine has become more greatly differentiated, and inequalities have been further exposed and aggravated. It is then clear that the poor, those in blue-collar and working-class occupations, women, and ethnic minorities have been disproportionally affected by the pandemic (Green et al. 2021; Romel 2020).

Attempts to predict the precise changes of the new global order are premature, but what we can say with some certainty is that the pandemic has revealed both locally and globally a series of worsening socio-economic inequalities and widening socio-economic and socio-cultural fault lines (Clark et al. 2020; Ferreira et al. 2021). The: "COVID-19 pandemic has disproportionately impacted the already marginalized groups in our societies" (Ali et al. 2020, p. 416) and societies have experienced ever greater economic, gender, racial, and age-based inequalities (Dang and Nguyen 2021; Deaton 2021).

One consequence is the calling into question, perhaps even the undermining, of certain hitherto taken-for-granted structures and ideals, among them acquisitive individualism, environmental destructiveness, and indifference toward the local community (Van Barneveld et al. 2020). The poor, including the working poor, have become less well-off and, at the same time, the gap between nations is widening, as economic disparities between countries grow, resulting in a contemporary economic and social world increasingly separated and segregated by the effects of the pandemic (Ahmed et al. 2022; Ali et al. 2020).

The pandemic has not only exposed inequalities within countries, but it has also brought into sharp focus differences between countries (Hilhorst and Mena 2021). The position of developing societies, with poorer, less established economies and often reliant on immature health systems ill-prepared to deal with the pandemic, has especially been exposed (Haldane et al. 2021; Shadmi et al. 2020) as socio-political divisions continue to be amplified by the pandemic (Muldoon et al. 2021). Many of these nations are also facing huge losses of revenue from tourism (Gössling et al. 2021) alongside collapsing export revenue and record levels of debt, causing the World Bank to suggest that the pandemic had led to a "tragic reversal" in development (Elliott 2021). There has been a move in some societies towards greater egalitarianism, with generalised beliefs that when it comes to wages and income and social equality, polities should reduce disparities and inequities. This may incorporate criticism of neo-liberalism and the free market (Child 2021; Gaynor

and Wilson 2020). We would anticipate seeing some of these suggestions and visions in the journalistic narratives we investigate.

### 2.3. The Interventionist State

While the pandemic cannot be seen to bring about more egalitarian political societies (in fact, circumstances might have worsened through the revelations of structured inequalities laid bare by the crisis), the impact of the changes that have been necessitated through a coordinated response have been equalising, sometimes deliberately so. Those in power have been faced with workers taking time off due to illness, and in fear for the continued stability and viability of regimes faced with the prospects of a dramatic decline in legitimacy and civility.

Under such conditions, the state has been left with little option but to extend welfare provisions and a range of benefits. Nowhere is this more apparent than in the realms of medicine, health, welfare, and social security. The required response—to ensure the good-enough health and wellbeing of all citizens in order to limit the spread of the virus and to buttress system integrity—has been to massively redistribute wealth and resources in order to avoid social disorder, widespread public suffering, and socio-economic disruption (Turner 2020).

Among the central object lessons of the global pandemic has been the widespread return to the strong and coordinated state as the only institutional structure capable of steering a rational and viable response to the emerging crisis. Countries have rapidly rediscovered the principles and practices of the Weberian legal-rational state, with an emphasis on meritocracy, professional staff, chains of command, bureaucratic coordination, the rule of law, a reduction in civil liberties, and compulsory orders, backed if necessary, by force. This is the interventionist state that steers political society from potential breakdown and disorder to an orderly and utility-maximising coherence.

The policy arena of greatest immediate importance in this respect is public health, and the pandemic has accentuated calls for greater equality of provision. As Vin Gupta (in Rasheed 2020) says: "People across the globe will use COVID-19 as a strong justification to demand universal healthcare". Global experiences of fear and insecurity and the pandemic-related worsening of conditions have conditioned many citizens to return to the coordinated interventionist state as a framework to furnish physical security, while deeper connections to ontological security have been shifting from faith in the charismatic leader, and exclusivist, essentialist conceptions of nation, religion, and gender toward selves that are secured through coordinated and expert leadership, and open, inclusive conceptions of an interconnected global order in which distinctions of nation, belief, and gender are less salient.

Certainly, a series of populist leaders and movements have attempted to re-invigorate essentialism, exclusivism, nativism, and racism (Agius et al. 2021). This includes the Canadian convoy protests, where the proximate rallying cry was against vaccine mandates themselves. However, such bids are decreasingly effective. Closing and reinforcing borders flies in the face of a pandemic which is clearly global in scope. Any rational solution to it will also need to be global in scope. The necessary curtailment of international travel might appear to re-enforce populist messages of wall-building and isolationism. Ironically, however, the Canadian protesters have built their case against vaccines on opening international borders, at least to those who refuse to declare their vaccination status.

There are calls for state intervention to protect and enhance the social wage through bolstering those parts of the economy essential to life, notably health, employment, housing, education, and basic income support. This involves a major cultural shift, placing a different set of values at the heart of the economy and society and reaffirming the dignity of socially necessary labour as well as care for those unable to work. State intervention is also called for to support the economy and social order. Widespread uncertainty, fear, and anxiety have reawakened the state as the more or less legitimate guarantor of law and order (British Academy 2021; Haug et al. 2020; International Monetary Fund 2022; MacFarlane 2021).

Such calls for an enhanced, or at least reinvigorated, state are anticipated as elements of the post-COVID-19 narratives we investigate in the legacy media sources.

### 2.4. The Local and Communitarian

Throughout the pandemic, we have seen a turn to the local, the familial, the communitarian, the known, and the familiar as locations of support. The footprint of the neighbourhood has widened and deepened. Exhortations at the municipal, provincial, and federal orders of the Canadian government have stressed the need to stay within walking distance of home, to support local businesses, and to celebrate the closeness of family and community bonds. The onset of second-order catastrophes in Canada, such as the Nova Scotia mass murders or the floods in northern Alberta, have redoubled community-affirming and supporting energies across the country. There has been a reaffirmation of the notion of community, resting on notions of solidarity and mutual aid. Here, the state takes a diminished or perhaps a complementary role. Individuals and small groups are recognised, affirmed, encouraged, and enabled to organise support and care within their local communities. The most ambitious projection of this for the future sees the emergence of renewed cultures of fraternity and solidarity (Alderden and Perez 2021; Alalouf-Hall and Grant-Poitras 2021; British Academy 2021; Christens 2012).

The implications of suddenly sending massive workforces of hundreds of millions of people from centralised offices, shops, and factories back into their homes, which are then more or less instantly retooled as remote and networked workplaces, are widespread and profound. This is not to mention the closure of schools and the quarantining of vulnerable populations in isolation. This is a radical acceleration of the twenty-first century version of putting out and homework as women in particular struggle to adapt to the new realities of juggling work and domestic labour.

At a minimum, this leads to a dramatic rethinking of the nature of work and the workplace, as well as work-life balances, notably childcare and responsibility for domestic labour. It also immediately raises issues of equity when it comes to access to the tools—bandwidth, software, and hardware devices—needed to conduct the multi-tasking work of the everyday in the physically distanced and separated world. The new normal carries with it all the advantages and disadvantages of domestic labour and dependent care, the convenience and flexibility as well as the humanisation of life-work balances, but also the breakdown of boundaries around work tasks, the 24/7 job, demands for availability, and the stigmatisation associated with opening the homes of poor and marginalised people to greater scrutiny.

As Peter Lunn notes: "This pandemic is far from a war, but it requires pulling together. And when people realise what collective action can achieve, it could change how they relate to others, resulting in a greater sense of community" (Rasheed 2020). The socio-political consequences of this are open and relate to the two sides of McLuhan's global village concept: the inclusionary and the exclusionary. It is certainly the case that by driving us increasingly to the familiar, the fine-grained, and the local, we enhance and strengthen our shared humanity, notably through the shared networks of cyberspace. In this sense, what is local and familiar may be generalised into a shared universal humanity. However, it is also plausible that current circumstances may drive some further toward parochialism and particularism, stigmatising, hounding, and excluding those deemed unfit or outsiders. The dark side of community and the separated family bubble also carries with it the threat of increased domestic stressors and violence, and there is evidence of a rise in rates of anti-women domestic violence (Gunraj and Howard 2020).

The threat of the global pandemic has dramatically restricted travel, notably airline travel but also, in many cases, local and regional travel. Apart from the obvious impact on the travel and tourism industries, each of which looks set to undergo dramatic transformation, there is a turn to local travel and tourism, from the footsteps that each of us may take from our places of residence as we visit local places to the development of the "staycation" industries in the future. This links with a growing environmental sensibility

that mega cruise liners and jumbo jets are environmentally unsound and are generating unsustainable problems for the planet. Associated with this is a forced rethinking of the food supply chain and perhaps an accelerated awareness of the importance of locally sourced food and plant-based food. Among the sources under consideration in our study, we anticipate references to the local and communitarian with respect to narratives of the post-COVID-19 polity.

*2.5. The Deeply Participative*

The role of the state and community animation has reawakened tendencies toward participative and deliberative democratic structures in society, with greater input of local societal and economic choices in decisions surrounding social spending and budgetary allocation and greater grass-roots democracy. (Child 2021; Greitens 2020; Mendes 2020; Saad-Filho 2021) In many ways, the responses to the global pandemic have conditioned a renewed and enriched participation in public life. At the very least, even minimal scanners have been obliged to "pay attention", if only for purposes of self-preservation. But there is an overall sense of involvement, commitment, and responsibility on the part of many. This has been conditioned by the widespread requirement for collective action for the general wellbeing of the community. Under such circumstances, each of us is called upon to be mindful, active, and engaged, if only to "do our bit" by staying in quarantine, maintaining physical separation, using face covers, or otherwise following the rules. We expect to see consideration of democratic structures and processes in the narrative sources investigated.

## 3. Methodology

In order to explore these views, our approach combines thematic analysis (Braun and Clarke 2006, 2012; Butler-Kisber 2018) with narrative analysis (Andrews 2013; Andrews et al. 2000; Andrews et al. 2018; Clandinin and Connelly 2000). Narratives are often stimulated or catalyzed at points of tension, stress, or trauma in order to make sense of rapidly changing or unstable lived experience. Narratives are constructed in history, social structural context, and in constant community with others through circulation, amplification, repetition, and dialogue. Thematic analysis generates a range of methods for sorting and classifying according to themes identified in text and talk. Following Braun and Clarke (2006), our thematic analysis blends our own theoretical expectations regarding post-COVID-19 polity narratives with an inductive approach to the data. Both inductive (data-driven) and deductive (theory-driven) characteristics of our textual data have informed the sorting, coding, and development of the themes.

To test our expectations and to assess how far they explore the themes and perspectives of narratives that were operational in the COVID-19 period, we ran a database search of the *Canadian Periodicals Index* (CPI) from 29 February 2020, to 31 July 2021. The CPI covers over 17 million articles from over 1500 Canadian newspapers, magazines, and journals in English and French. This includes some Canadian editions of foreign-based publications. About 800 of the periodicals are available in full text from 1983 to the present. The emphasis is on current events, culture, the arts, technology, business, and commentary.

We searched a series of terms across the CPI search engine, using a keyword search, related to "Covid", "Pandemic", and "Coronavirus". The terms are listed in Table 1 below. We then scanned each of the 2227 full-text articles that were generated from that search from news sources, magazines, and academic journals, excluding those that were duplicates, fewer than 300 words in length, and those written in French. From the initial output of 2227 articles, we also excluded those that were overtly and clearly not related to narratives concerning the post-COVID-19 polity. This resulted in a final corpus of 228 articles, or just over 10 percent of the original number. Using the discourse analytical software programme, NVivo, we then coded the 228 articles into a series of nodes, which are presented in Table 2 below. The resulting nodes generated the illustrative narrative materials that we draw upon in the substantive sections of the article. As we shall demonstrate, our anticipated narrative themes were generally present across the sources, thereby supporting our theory-

driven deductive expectations. The most striking inductive (data-driven) outcomes of the study were the distinctions between predictive and prescriptive content among the narrative materials describing the post-COVID-19 polity. In our original theorisation of post-COVID19 narratives, we had not anticipated this distinction. Scrutiny of the data made it apparent, and we decided to code according to the distinction. Case 86 (see the Data Sources for a key to numbered sources in the database) illustrates a predictive narrative:

> The crisis has also revealed government's ability to provide solutions, drawing on collective resources in the process. A lingering sense of "alone together" *could* boost social solidarity and drive the development of more generous social protection down the road.

> Case 105 demonstrates a prescriptive narrative approach:

> For years social spending has favoured the elderly and an outdated safety net. It *should* be rebuilt around active labour-market policies that use technology to help everyone.

The columns of Table 2 specify both sources (individual articles) and references (extracts of text from articles that express predictions and/or prescriptions related to our study). The number of sources exceeds 228 because some articles had multiple codes into a range of themes. Predictions and prescriptions that did not describe the post-COVID-19 polity were coded into the generic category "Other". For example, several sources concerned travel and tourism trends/destinations or business sales trends and preferences after the pandemic. While of interest, these were not of direct relevance to our study.

**Table 1.** Canadian Periodicals Index Keyword Search (29 February 2020–31 July 2021).

| SEARCH TERM | NEWS MEDIA | MAGAZINE | ACADEMIC JOURNAL |
|---|---|---|---|
| **Post Covid** | 1001 | 103 | 4 |
| **Post Pandemic** | 887 | 121 | 15 |
| **Post Coronavirus** | 51 | 2 | 3 |
| **Life After Covid** | 17 | 3 | 1 |
| **Life After the Pandemic** | 4 | 1 | 0 |
| **Life After Coronavirus** | 2 | 1 | 0 |
| **The World After Covid** | 2 | 4 | 0 |
| **The World After the Pandemic** | 0 | 5 | 0 |
| **TOTALS [GRAND TOTAL = 2227]** | **1964** | **240** | **23** |

According to the data in Table 1, the mainstream Canadian news media contained substantial materials on the post-COVID-19 polity, and that the term "Covid" was preferred over the earlier term "Coronavirus", which has now fallen out of use. As can be seen, the term "pandemic" was a close second in terms of references to the potential for future developments. It is also understandable that attention to the post-COVID-19 polity was limited in academic sources, as academic research typically takes longer to conduct, peer review, and publish.

The data in Table 2 permits an initial examination of the extent to which the narratives of the post-COVID-19 polity found in the CPI search correspond to the theoretical or deductive (Braun and Clarke 2006) themes we identified earlier. As illustrated earlier, following our inductive thematic analyses, we observe that conceptualisations of the post-COVID-19 polity fall into two broad categories: those that predict or speculate about the future, often in neutral terms, and those that take a stand and make prescriptive judgments concerning the future.

**Table 2.** Nodes and Number of References Derived from Sources (NVivo).

| NAME | SOURCES | REFERENCES | SOURCES | REFERENCES |
|------|---------|------------|---------|------------|
| | PREDICTIVE | PREDICTIVE * | PRESCRIPTIVE | PRESCRIPTIVE * |
| Rationality and Science | 6 | 7 (87.5) [2.4] | 1 | 1 (12.5) [0.7] |
| Social Equality | 18 | 24 (43.6) [8.2] | 20 | 31 (56.4) [22.5] |
| Wage Equity and Guaranteed Income | 3 | 3 (21.4) [1] | 9 | 11 (78.6) [8] |
| Interventionist State—Economy | 9 | 14 (53.8) [4.8] | 9 | 12 (46.2) [8.7] |
| Interventionist State—Order | 8 | 10 (76.9) [3.4] | 2 | 3 (23.1) [2.2] |
| Interventionist State—Welfare State | 10 | 11 (26.2) [3.8] | 18 | 31 (73.8) [22.5] |
| Local and Community | 31 | 44 (78.6) [15] | 6 | 12 (21.4) [8.7] |
| Local Sourcing and Craft Production | 15 | 18 (81.8) [6.1] | 3 | 4 (18.2) [2.9] |
| Deep Democracy and Participation | 4 | 8 (100) [2.7] | 0 | 0 (0) [0] |
| Environmental | 8 | 16 (61.5) [5.5] | 9 | 10 (38.5) [7.2] |
| Global and Cosmopolitan | 9 | 13 (61.9) [4.4] | 6 | 8 (38.1) [5.8] |
| Spiritual/Philosophical | 9 | 14 (51.9) [4.8] | 10 | 13 (48.1) [9.4] |
| Other | 99 | 111 (98.2) [37.9] | 2 | 2 (1.8) [1.4] |

* Row percentages for references given in round brackets; column percentages for references given in square brackets.

As is clear in Table 2, most of the references (over 55 percent) are predictive, but there are also a substantial number of narratives that are prescriptive. When it comes to prescriptive narratives, our classification and that of CPI contributors is close. Among the data, one feature that stands out is the dominance of the prescriptive voice used in narratives concerning broadly social justice issues, notably wage equity and the desirability of the welfare state. Conversely, those narratives focused on community and working from home tend toward the predictive end, and there are relatively fewer prescriptive narratives in these categories.

Using the number and percentage of references as our guide it is clear that in general terms, the journalists and commentators writing across Canadian periodicals have conceptualised the post-COVID-19 polity according to similar criteria as those we outlined. Specifically, when it comes to the prescriptive there are few that we did not also consider. A rough guide to the degree of correspondence with our initial five themes is attainable by summing the number of predictive and prescriptive references in each thematic area and then expressing them as a percentage. As the two references columns indicate, there is a total of 431 references. In terms of our initial themes, the percentage correspondences are as follows: (1) rationality and science (1.9 percent), (2) social equality (16 percent), (3) the interventionist state (18.8 percent), (4) the local and communitarian (18.1 percent), and (5) the deeply participative (1.9 percent). The remaining substantively coded themes (Environment, Global and Cosmopolitan, and Spiritual/Philosophical) make up 17.2 percent, and the other category makes up 26.2 percent.

Of the five deductive or theoretical themes we conceptualised, three receive substantial coverage in the post-COVID-19 narratives derived from the CPI search. These are: social equality, the interventionist state, and the local and communitarian. There are very few references to either rationality and science or the deeply participative. One possible explanation for this is that such concepts and themes are relatively theoretical and abstract and, thereby, do not lend themselves as readily to journalistic coverage. Of the few references in these categories, almost all of them come from two of the more serious political periodicals, the Canadian edition of *The Economist* and *Foreign Affairs*, which are also referenced in the CPI.

The remainder of the article now turns to a discussion of our findings with respect to each of the five themes.

## 4. Discussion

When it comes to the issues of rationality and science, as noted earlier, they do not appear to be very prominent in the narratives regarding the post-COVID-19 polity. Of the small number of references to rationality and science, Frances Fukuyama in Case 86 adopts a similar view to our own: "The practical realities of handling the pandemic favor professionalism and expertise; demagoguery and incompetence are readily exposed. This should ultimately create a beneficial selection effect, rewarding politicians and governments that do well and penalizing those that do poorly" (Case 86). In Fukuyama's post-COVID polity-19, rational choices in the political marketplace lead to optimal public policy choices.

Others conceptualising this theme refer to connections between recognition of the pandemic and support for climate change measures, the role of churches in combatting disinformation, the role of technology in post-COVID-19 polities, and what Bratton (2021) calls "the revenge of the real". The following prescription by Danielle Allen brings together an aspiration for scientific rationality with a call for civic literacy:

> The United States needs science. It needs technological innovation, and it needs scientists to advise elected leaders. But that is not all the country needs. It also needs people who can interpret the science and make judgment calls that take broader factors into account. The U.S. government's growing investments in scientific education have been accompanied by reductions in funding for civics education . . . And the country is paying for it now. In the United States today, the art of governance is, at best, on life support. Paradoxically, Trump has delivered the best civics lesson in generations. Thanks to his impeachment trial, Americans have had to think about the proper bounds of executive power, the checks offered by the legislative and judicial branches, and precepts of the Constitution. Thanks to his failure to govern through this crisis, many have learned for the first time just how the United States' federal system is supposed to work. If the country's constitutional democracy is to have a healthy future, Americans should finish this crisis intending not only to invest in health infrastructure but also to revive civics education. (Case 88)

This narrative of a post-COVID-19 future places science and rationality at the centre of expectations for the American polity and offers the vision of a society that is only healthy with the necessary investments in both the natural and social sciences.

As detailed earlier, the crisis has revealed the perceived injustices and inefficiencies of low wage levels for critical workers, many of whom are racialized minorities and women. The concept of a universal basic income has made a comeback into public discourse (Maclean's 2020), alongside more generalised calls for greater income and wealth equality. Predictions of and calls for greater socio-economic equality as well as wage equity are prominent among the references listed in Table 2 and constitute a dominant theme among contributors to public deliberations as they envisage the post-COVID-19 polity. Hopes and expectations regarding wage equity are for a universal basic income, economic dignity, notably for front-line workers, reductions in disparities in income and wealth, better bargaining rights, and improvements in working conditions in a post-COVID-19 polity.

When it comes to social equality, the narrative threads incorporate racialised, gender, and more general disparities and the opportunity to address them in a post-COVID-19 world. This includes calls for the intergenerational redistribution of wealth, improvements for front-line workers, improvements in housing, health care, food security, and access to the arts, culture, and education, including access to digital resources.

As illustrated in Case 203, Toronto author and social justice activist, Rusul Alrubail, and her colleagues develop prescriptive narratives of improvements for marginalised workers. As they argue, the pandemic has turned the spotlight on a number of important issues facing workers, such as inadequate health and safety conditions, the alarming rise of precarious work, and the need for a living wage. These issues disproportionately affect vulnerable workers, including women, racialized people, and other equity-deserving groups that comprise a high percentage of the labour force in some of the hardest-hit sectors (Case 203).

The predictive narratives also contain warnings about the post-COVID-19 polity should certain trends continue or grow. These include the challenges of inflation and the cost of living, the failure to address the working conditions of precarious workers, and growing socio-economic inequality. In a further illustration of these themes, this extract envisions the nature of economic dignity in a post-COVID-19 polity, regarding it as the attainment of material sufficiency and the absence of desperation, the wherewithal to build lives that go beyond survival, which incorporate self-actualisation, and the capacity to freely engage as a worker, consumer, and citizen:

> Gene Sperling, economic policy chief under Clinton and Obama, says economic dignity should be "the ultimate goal of [post-Covid] economic policy". He says it [consists of] the following three pillars: "The ability to care for family without economic deprivation or desperation"; "the capacity to pursue potential and a sense of purpose and meaning"; and "the ability to contribute and participate in the economy with respect, free from domination and humiliation". (Case 153)

Taylor's (2021) prescriptive narrative in Case 181 gestures to a post-COVID-19 world in which wages, working conditions, and dignity for front-line workers will have benefitted from a diminution in the gap between rich and poor.

> I'd rather talk about how we're continuing to shortchange vital workers while Canada's wealthiest CEOs are earning record incomes. Surely our dreams of post-pandemic work should address this widening gap? As it stands, dignity, protection and decent wages for front-line workers are still an afterthought. (Case 181)

As revealed in Table 2, above, there are multiple instances of both predictive and prescriptive narratives regarding the interventionist state, with respect to steering the economy, supporting, or expanding the welfare state, and, to a lesser extent, sustaining order. The general trend of narratives concerning the economy is related to how states will deal with the vast accumulated deficits incurred during the pandemic. One direction is

continued stimulus spending on the basis of activism and big government. As Ned Temko says: "Now the authorities everywhere are using state spending to prime the postpandemic economic pump" (Case 175). Specific mention is made of infrastructure spending, the green economy, support for business recovery, education and skills training, as well as large-scale public sector job creation in public works projects. Related to this are calls for better working conditions and equity for women and racialised minorities. Concerns are expressed about the size of the deficit and what fiscal measures might be needed to cope, including raised taxes and cuts to programs.

Narratives range in specific details, but many call for bigger state spending in general as well as using the pandemic as an opportunity to rethink priorities on a grand scale. Temko illustrates the point by sketching the post-COVID-19 intentions for the interventionist state among moderate-to-conservative leaders:

> Big government is staging a dramatic comeback, and U.S. President Joe Biden's $2.6 trillion infrastructure and investment plan is just the latest sign. After all, his predecessor, Donald Trump, signed off on nearly double that amount in pandemic spending. Across the Atlantic, the government of British Prime Minister Boris Johnson has broken with the orthodoxies of Ms. Thatcher and his other Conservative predecessors to announce billions of dollars in new spending, along with higher taxes to pay for it all. German Chancellor Angela Merkel—once the very embodiment of fiscal restraint—has signed off on stimulus and recovery plans to the tune of nearly $1.5 trillion. She's also embraced the idea that an activist government should support and even buy into companies critical to Germany's future economic strength. (Case 175)

Among the most comprehensive and detailed narratives are those calling for a renewal of the Canadian welfare state in the post-COVID-19 setting. Reference is made to the spirit of the New Deal and a return to the Keynesian concept of government as a solution rather than a problem. There are a few specific critiques of capitalism and neoliberalism, but most prescriptions are more generalised and less overtly ideological. There are many specific calls for particular programmes or initiatives in health, long-term care, anti-poverty initiatives, education, the environment, social equity, senior care, universal basic income or guaranteed income, childcare, housing, and other services. This envisaging of a green and sustainable social infrastructure typifies the tone of the 31 prescriptive references regarding the welfare state (see Table 2):

> Public policy and finance are understandably focused on resilient recovery and rebuilding, with unprecedented investments in physical infrastructure to create green jobs that address the imbalance between humanity and the natural environment. To fulfil the potential of this great transition, our social infrastructure needs attention too. "Social infrastructure" includes policies, practices and relationships that enable us to create a more resilient, inclusive and sustainable society, from the grassroots to the global, and spanning health care, education, culture and our democratic processes. COVID-19 has revealed systemic injustices and vulnerabilities including institutionalized racism, substandard seniors housing, the lack of paid sick leave and inadequate childcare. To this list we can add Indigenous reconciliation, the income and wealth gap and the life- and budget-sapping increases in chronic disease. (Case 207)

There is a generalised concern for an orderly polity, but not many references to the role of the state in maintaining order. Given the phasing of the pandemic and the high degree of societal cooperation throughout the period under investigation in our study, this is understandable. In light of the more recent attempted sedition/insurgency of the Canada convoy protests and its blockages and occupations, as well as the use of emergency powers in response by the federal state, there might now be a greater degree of attention to order and the interventionist state in the post-COVID-19 polity. Of the themes developed across narratives of the state and order in the post-COVID-19 polity, there are references both to

the potential for social unrest and political upheaval and to the need to revitalize military resources and personnel as well as atrophied national and global institutions in order to deal with crises and emergencies. While some express concern for the future surveillance apparatus, others, such as this British narrative from The Economist, see in the military an opportunity to build reserves for civil emergency support and to generate economic and trading opportunities:

> Penny Mordaunt, a minister responsible for civil contingencies and co-author of "Greater: Britain after the storm", wants the state to harness those who volunteered to battle coronavirus, directing them towards "national missions", such as elderly care. Ministers plan to overhaul military reserves, and create a new cadre of civilian reservists, such as retired doctors and civil servants, who can be mobilised in crises. (Case 4)

If we return to Table 2, local and community themes and narratives are prominent among the references made to post-COVID-19 polities in the sources we explored and typify many responses. It is clear, however, that the general approach is predictive rather than prescriptive. Specific attention is focused on the opportunity to develop the working from home concept, with its associated risks and opportunities. There is a strong sense of inevitability that the work-life balance of the future will incorporate greater flexibility regarding working from home as well as more options regarding flexitime. While these developments are generally presented as progressive, a few references point out the challenges of equity in working from home. This description of one PR firm expresses the ways in which a progressive narrative of diversity, inclusion, and care is blended into the new business model of hybrid and flexible workplaces, workspaces, and work schedules:

> Weber Shandwick is shifting to a permanent hybrid model (once it is safe for people to return to the office), with employees going in three days a week and working remotely the other two. This model, says Gail Heimann, president and CEO of the public relations giant, allows for a more inclusive and diverse organization, as well as helping to foster a better work-life balance for employees. By providing flexibility to work from various locations, it opens up opportunities to hire more diverse people who typically wouldn't have been considered, like those who don't live near a major metropolitan area, Heimann says. As the ad world takes a hard look internally at its horrific history with diversity and inclusion, this ability to think differently about where they can hire from presents an opportunity to expand the talent pool ... Amid a crises of women leaving the workforce during the pandemic due to the demands of trying to balance careers alongside childcare (including virtual schooling) and other caregiving responsibilities, a hybrid model can help empower women and other groups without compromising family life. (Case 97)

Other contributions to the themes of local and community as well as local sourcing and craft production stress the possibilities inherent in a return to devolved and outsourced businesses in the community, with cafes, bookstores, libraries, churches, and other entities serving as community hubs. There are references to post-COVID-19 domestic and local tourism, fewer business trips, and less commuting. The virtues of local restaurants, food supply chains, transit options, pedestrianised streets, and shared public spaces are promoted in a post-COVID-19 society. Specific reference is made to local music and arts production, home gyms, the creation of smaller and more intimate care homes for seniors, the upsurge in online shopping, and home renovations, each of which is predicted to continue into post-COVID-19 communities.

There is relatively little direct evidence among the CPI of references to deep democracy or participation in the political sense. Of all the categories, this is the one that is exclusively predictive (see Table 2) and also open with regard to the predictions made. In brief, most references express the precariousness of democracy and its vulnerability to pandemic circumstances. Will there be a political backlash? Can democracy survive following

the experience of control and regulation? Will unscrupulous leaders take advantage of the circumstances? These two extracts from an opinion piece by Francis Fukuyama are illustrative of the narratives of the future. As he says, there are alternative and competing visions of the future:

> The practical realities of handling the pandemic favor professionalism and expertise; demagoguery and incompetence are readily exposed. This should ultimately create a beneficial selection effect, rewarding politicians and governments that do well and penalizing those that do poorly. Brazil's Jair Bolsonaro, who has steadily hollowed out his country's democratic institutions in recent years, tried to bluff his way through the crisis and is now floundering and presiding over a health disaster. Russia's Vladimir Putin tried to play down the importance of the pandemic at first, then claimed that Russia had it under control, and will have to change his tune yet again as COVID-19 spreads throughout the country. Putin's legitimacy was already weakening before the crisis, and that process may have accelerated. (Case 86)

> Over the years to come, the pandemic could lead to the United States' relative decline, the continued erosion of the liberal international order, and a resurgence of fascism around the globe. It could also lead to a rebirth of liberal democracy, a system that has confounded skeptics many times, showing remarkable powers of resilience and renewal. Elements of both visions will emerge, in different places. Unfortunately, unless current trends change dramatically, the general forecast is gloomy. (Case 86)

Table 2 indicates four themes that fall outside the five original theoretical themes we developed. These classifications were inductively developed from the data and constitute a range of themes that we had not originally conceptualised. We offer some initial and preliminary analysis of these themes but acknowledge that fuller exploration of them is beyond the immediate scope of this article. A substantial category of themes unrelated to our study is labelled as "Other", and they are almost exclusively predictive. These include a large number of business and tourism trends that writers believe will characterise post-COVID-19 societies. Three other categories were of sufficient interest to us to classify them separately in the Table. Related to the other themes, notably local and community perspectives, the interventionist state, and rationality and science, the environment and sustainability emerged in both predictive and prescriptive narratives. Margaret Atwood details her vision for a green and sustainable planet:

> I hope that our desire to green the planet will become stronger post-COVID. Many have turned to Nature during this period, realizing for the first time that it is part of them. Every breath you take contains oxygen made primarily by marine algaes. Kill the oceans and we're dead. My hope is that most people and countries will finally realize that, and take the necessary steps. (Case 217)

> So, let's hope there will many new kinds of jobs. Plastics will have to be rethought so they are less destructive, and cleaning up the plastic mess that's already out there will take years. Lots of jobs there! New uses for unexpected materials are already coming online—heard about the mushroom coffins? I hope that energy conservation will become a widespread goal, and less polluting energy sources such as hydrogen will be deeply explored. (Case 217)

A number of sources aspire similarly to forms of global, cosmopolitan, and multilateral connectivity, a united approach that emerges from the shared challenges of confronting the pandemic itself, the artificiality of borders, and the limitations of national interests. Those constructing narratives of global cooperation express the hope that the spirit of connectedness throughout the pandemic will expand into a deeper and renewed sense of global community in the post-COVID-19 era.

The final theme is labelled "Spiritual/Philosophical" and, unsurprisingly, many of the sources associated with it are religious, denominational, or church-based publications. It would be rash to make too many claims regarding the role and place of organised religion in the current pandemic. However, enforced isolation and distancing has certainly enhanced religiosity in some, even among atheists and agnostics, a quest for some spiritual or philosophical meaning and guidance. Signs of growing spiritual searching, religious observance, and quests for meaning are evident in the output of many of those affected by the global pandemic and condition many of the more spiritual post-COVID-19 narratives. As Bryan Turner says: "Perhaps it is unsurprising, therefore, that together with natural disasters, they have evoked religious responses to calamity that bring into question the meaning of life and its injustice" (Turner 2020). Kofi Hope's vision of "building back better" in the post-COVID-19 world expresses the theme of spiritual connection and its relevance to changing the world for the better:

> We must cut through the culture wars and echo chambers to build consensus on the parts of society needing significant reform. We also need to subvert the ingrained cynicism of our modern age and rebuild our confidence that change is possible, that we can achieve great things together. Many leaders, thinkers and writers are taking on these challenges, but I believe there is one tool in the social change toolbox that we've neglected to use. Spirituality . . . for thousands of years, spiritual approaches to framing and making sense of our world have been used in times of crisis for sense-making, for having collective conversations on our challenges and collective aspirations. The stories and philosophical insights of our religious traditions, along with personal accounts of spiritual learnings—all can help move conversations on big issues from facts and figures to the level of values. (Case 208)

## 5. Conclusions

It is appropriate that the final narrative thread considered is that of a man called Hope. A common perspective across the narratives found in the sources and references considered in our study is a sense of hope, particularly in those that are prescriptive of the future. Sustaining that hope is a widespread faith in the propensity of individuals and communities to work toward the general improvement of shared conditions and circumstances. Significantly, Capelos and Demertzis characterize individuals in ressentiment as lacking hope "because of their powerlessness and self-victimization" (Capelos and Demertzis 2022, p. 5). The emphasis on hope and faith is not surprising, in that the criteria for inclusion in the study were a reference to the future and a post-COVID-19 or post-pandemic polity. Of course, a view of the future may be dark, even despairing, and hopeless. However, as our data demonstrates, such perspectives did not characterize the 228 sources in our study. Generally, corresponding to our initial theorization of perspectives on the post-COVID-19 polity, the journalists and commentators whose articles we used in our data analyses wrote substantially on narratives of social equality, a renewed interventionist state, and the importance of the local and community. This was particularly so for those references that we classified as prescriptive. The predictive/prescriptive distinction emerged from our reading of the data and reminds us both of the preliminary nature of our research into narratives of the post-COVID-19 polity and the importance of keeping our theoretical frames open to the themes of the empirical data we explore.

As we further noted, there were themes that remained to be developed further, specifically those that we had not originally theorised. Further analyses might pay greater attention to differences among publications according to the phasing of their date of publication. While we used the narrative form in our analysis, it was more of a survey than a fine-grained exploration of the narratives themselves, and it would be interesting to investigate their deeper meaning and implications. We did not differentiate media sources according to their ideological leanings, and this comparison of core belief systems across the media could produce a more refined set of findings. So too would a companion study

of right-wing and populist media sources. We should add that among the large corpus of data we investigated, including those themes originally conceptualised in our theoretical considerations, there remains a broad range of further questions to investigate. In the fast-emerging worlds that have arisen from the pandemic, no singular schema or set of expectations will be sufficient to capture the complexities. Each of us must continue to pay attention, revisit our ideas and the data in the context of new developments, and learn.

The fault lines of the current era are drawn by the closed reactionary goals of the sources of information that inform the Canadian convoy protests. These centre around the ultra-right-wing Fox News and algorithmically-driven social media sources of disinformation and conspiracy theories. The mindset of the protesters is a return to the past, conceived as simpler, purer, and fairer. To this end, the current regime is regarded as having deprived them of their rights and freedoms in an oppressive manner. The movement combines anti-vaccination activists with white nationalists, members of the alt-right, and those standing against "elites" and "political correctness". In contradistinction to this, the progressivism of the mainstream of Canadian journalism, which we have examined through the CPI search engine, is professionally neutral in terms of its set journalistic standards and may include broadly progressive and hopeful editorialising with regard to expectations of the post-COVID-19 polity.

A global pandemic is a world-changing event, altering lives, and seemingly assured social realities in a period of global uncertainty and crisis. At the time of writing, in mid-2022, it is too soon to know the depth and breadth of changes that will emerge from the current circumstances. However, if we and those writing across the Canadian legacy media are right about what has been happening with rationality and science, social equality, the role of the interventionist state, the local and the communitarian, and deep participation, the political changes brought about by the pandemic are likely to be lasting and notable.

But, of course, in light of the shock of the uprising of the Canadian convoy protests and other similar events in other places, none of these claims are straightforward or immediate. In fact, there are a series of acute struggles for hegemony with respect to both material claims and ideational ones. It was Antonio Gramsci, in his *Prison Notebooks* (Gramsci 1971), who pointed to the dangers of an old order dying when a new one could not be born. In these circumstances, the world becomes a much more dangerous place, with an increase towards primary, even irreconcilable antagonisms. Much of the struggle is not just with the virus itself, but also with how this period is narrated and whether it is presented as a series of war stories or whether different narratives involving societal change, agency, and different futures can gain traction.

At the end of the contemporary crisis, many will seek to offer alternatives to the return to the old norms and business as usual, or, even further, to recover mythologised pasts. It will be interesting to see how far the voices we have identified remain dominant narratives, able to force an open debate around the exit strategies from the pandemic. The question of whether such narratives can become embedded or lead to more inclusive, equal, and democratic societies rest largely on the role of civil society.

It will be years, perhaps even decades, until the vast social, political, economic, and technological outcomes brought about by the pandemic are fully realized. Those anticipating a return to pre-pandemic normality may be shocked to find that many of the previous systems, structures, norms, markets, and employment are no longer there to return to. The extracts we have presented represent the predictions as well as the prescriptions of the many journalists and commentators who share our interest and prefigure the future in their narratives. In the words of The Whos' "Won't Get Fooled Again", it may be some time until we know if the "new boss" is the same as the "old boss".

**Author Contributions:** Conceptualization, P.W.N.-L.; Formal analysis, J.W.M. and P.W.N.-L.; Writing—original draft, J.W.M. and P.W.N.-L.; Writing—review & editing, J.W.M. and P.W.N.-L. All authors have read and agreed to the published version of the manuscript.

**Funding:** This research received no external funding.



**Institutional Review Board Statement:** No Institutional Review Statement.

**Informed Consent Statement:** Not applicable.

**Data Availability Statement:** Data available publicly through the Canadian Periodicals Index.

**Conflicts of Interest:** The authors declare no conflict of interest.

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
