# Peer review of "Imagining the Post-COVID-19 Polity: Narratives of Possible Futures"

_socsci, doi:10.3390/socsci11080346_

Round 1
Reviewer 1 Report
I read this article with great interest as it addresses a very interesting topic. The article details the forward looking writings of mainstream journalists and opinion leaders on visions of post-covid polity. It uses thematic analysis and narrative analysis to show how rationality and science, social equality, notions of interventionist state, local and communitarian politics, and deep participation are discussed and imagined in the selected texts. The authors conclude that the majority of their sources called into question structures and ideas that were taken for granted prior to the pandemic providing a counter-narrative to the reactionary voices of the Ottawa occupier movement.
By analysing future imaginations, this article opens a critical space to examine and discuss visions for new ways of thinking about social and political life. In this sense it provides a thorough account of progressive (and at times radical - although not identified as such) visions of the future.
I think with some improvements this article will make a significant contribution to Social Sciences, so my feedback focuses on suggestions for improvement in the order I encountered material in the text:
ABSTRACT: The Ottawa occupiers are central to contextualizing the analysis but are not listed in the abstract. I think this is an omission that should be addressed since their reactionary vision contrasts the material analysed by the authors.
INTRODUCTION: It would be helpful to start the introduction with the research aim of the manuscript, so the reader situates the discussion of the following topics with this in mind.
Clarify the term ‘legacy journalists’ for the readers.
STATE OF THE ART: The review of the state of the art literature is too short because relevant academic discussions are dispersed through the manuscript – this makes the manuscript appear thin on theory, while actually it is not. I think this presents an opportunity to consolidate theoretical and academic debates in one dedicated section and thus enhance the theoretical contribution of the manuscript. I think the authors should add:
- A discussion of reactionary vs progressive visions of politics, adding references of theoretical debates in the field (see for example the open access special issue on Reactionary Politics and Resentful Affect in Populist Times in Politics and Governance (https://www.cogitatiopress.com/politicsandgovernance/article/view/4727) or the special issue on Innovation (2022) on Affective Responses to Crises in Europe https://www.tandfonline.com/toc/ciej20/current
- A discussion of comparable contemporary movements to the Ottawa occupiers in the USA and other western democracies,
- A dedicated discussion of the topics discussed in the methodology: rationality and science, social inequality, the interventionist state, the local and communitarian, and participation. These discussions now appear in the methodology (p3) and also in the analysis section, but should be moved to the theory section to provide the theoretical spine of the manuscript.
METHODOLOGY: The methodological choices are sound and appropriate. The authors examined 228 articles across news media, magazines and academic journals that contained predictions and prescriptions concerning the post-covid polity. The methodology section can be revised to make the details of the empirical steps of the study clearer. For example, the authors should provide more details on the steps of their narrative/thematic analysis; clarify how the prescriptive vs predictive judgements were identified in the data – the keywords or meanings used to distinguish these categories; clarify which were the keywords used to identify each of the five themes; clarify the criteria for reactionary/progressive content (a very interesting distinction which is now mentioned briefly in the conclusion).
Regarding the selection of sources, the authors should clarify whether any of the academic articles that are treated as data are also those discussed in the literature review that define the hypotheses of the study. This would be problematic as the same sources cannot be used to draw hypotheses and test them. I do not think the authors made this mistake, but it would be good to clarify it here.
RESULTS: The thematic discussion in the analysis section offers interesting results – I think the authors can do more to clarify and enhance the value of their findings. One essential step to this direction is to separate the narrative material from the theoretical debates that are presented alongside it. To be more precise, material now presented in the results section fits better in the theory section: rationality and science on p.8, social equality p.9, interventionist state p.10, local and communitarian p 12, deep participation, p14. as it would allow the authors to build their expectations – which are then tested by their empirical project. Once these discussions of the state of the art move to the theory section, the analysis section can focus on the findings of the narrative analysis of the examined documents. The discussion/conclusion section can then reflect on the links between the findings of this project and the findings of extant studies in the field.
I also recommend the authors present more systematically the prescriptive and predictive material in each of the five thematic sections, by using subheadings or dedicated paragraphs.
The presentation of the narrative material in text becomes a confusing by using Author and Year of Publication. It would be clearer to present these quotes by case number in text and then follow with the reference details – or the reference details of the relevant cases can be provided in an appendix.
It would be interesting to see whether the data analysis shows any differences in narrative content when the authors take into account the political orientation of the media sources included in the sample. While most material is coded as progressive, a discussion of the ideological orientation of the sources (left/right/mixed/independent) would be helpful here.
CONCLUSION: The discussion on the sense of hope in the conclusion is very interesting but underdeveloped, particularly by placing it in the conclusion section. The authors can fold this into the theoretical section and preface the findings section with the hope analysis. It was useful to read the breakdown of progressive vs retrogressive coding of the texts (p.16) and I think this would have been very useful at the top of the analysis section as it helps contextualise the key findings.
The discussion on the first paragraph of p.16 can be expanded to offer more insights on reactionary vs progressive voices in Canada and link to extant studies in the field.
The authors should add a discussion of limitations for this project (methodological or other) and expand the ideas for extension for future research. It would be interesting to examine counter-narratives, if any. And as the authors note, it would be very interesting to see how these progressive calls materialize in the future – with a follow up project perhaps of coding how many of these visions are implemented (p.16).
Author Response
IMAGINING THE POST-COVID POLITY: NARRATIVES OF POSSIBLE FUTURES
RESPONSE TO REVIEWERS
Thank you to the reviewers for their thoughtful and helpful commentary on the manuscript. As they will see, their central comments were most useful to us in our thoroughgoing revision of the manuscript, and we applied their ideas fully and completely. While we remain responsible for the analyses, we believe that the reviewers have helped us greatly in developing a sharper and more focussed article.
Reviewer #1:
- The Canadian Convoy protests will now be mentioned in the Abstract, as requested by the reviewer
- The Introduction will now include a research aim statement
- ‘Legacy journalist’ will be explained briefly
- The reviewer states that the “review of the state of the art literature is too short”. In fact, there is no section of this kind at present, and it was not intended to be in the article in this way. In order to meet the request of the reviewer, with which we agree, there will now be a section to be called “Theoretical Perspectives.” This section will include a discussion of reactionary v. progressive visions and will move the theoretical materials currently in the methodology and substantive sections to the new theoretical section.
- The Methodology section will be revised and further specified as per the request of the reviewer, and the academic sources will be differentiated from the data base
- As stated earlier, the theoretical elements in the Results section will be placed in the new Theory section. As requested, the revised manuscript will now differentiate more clearly in the analysis of the prescriptive and predictive materials, most importantly highlighting that the distinction itself is inductive and data-based – a point understated in our original version. The narrative materials will now be identified by case number and a reference list in the Appendix. This should also clarify the distinction between academic sources and data, discussed earlier. The manuscript already contains some references to the ideological slant of the sources. These will be made more explicit, as requested by the reviewer.
- As requested, the data on progressive v retrogressive coding will be moved to the analysis section. As requested, the article will now make links to studies on reactionary v progressive voices. The section on research limitations will be expanded.
Reviewer #2:
- In response to the reviewer’s request, the entire manuscript has been redrafted to improve coherence and the reviewer’s requests for further evidence have been addressed. The manuscript now adds “9/11 and the War on Terror” as a defining event in the contemporary era.
- Following the reviewer, we have explained the national scope of our article and the Canadian context of our analyses. We believe that this further explanation will enable readers to appreciate the distinctions we draw re. the CPI as an index of journalism and commentary in Canada, but not necessarily from Canadian sources and linking Canada increasingly to a global world
- The manuscript now contains an explanation of predictive v. prescriptive threads, as requested by the reviewer. Also, in general, there is a more elaborated discussion of the qualitative analysis. The reviewer asks a good question regarding the underdevelopment of the unanticipated themes. The manuscript now addresses these issues.

Reviewer 2 Report
The manuscript explores visions of a post-covid world as presented in the Canadian mainstream press during March 20220 till August 2021. The authors use a database keyword search to locate articles, then subject them to an inductive and deductive TA. The manuscript is interesting, but suffers some limitations which would need to be revised before publication.
Firstly, the manuscript should be rewritten so that is more coherent, fully evidenced, better signposted and less journalistic in style. For example, the authors need to support all their claims (see lines 24-30 for example.) While I agree with you that covid is a defining event and the Ukraine War has the potential to be another, there have been other defining events in the 21st century (e.g., 9/11 and the resulting 20-year War or terror) which had profound impacts on the globe, try to support these claims, they may be more contested than at first cite. Proof read and tighten up the academic writing (e.g., lines 100-104) you could just have included the references, why include the information which repeats your original claims? Also, below (lines 108-111) you discuss a growing gap between nations, then repeat the claim again on 112-113. In places the writing is very journalist and one sided (for example lines 412-413). This is an academic article, it should be more balanced and evidenced.
The aims and context also need to be more focused, why explore the CPI? Is the manuscript and research only focus on Canada, North American, the West or the globe, it isn’t clear what your parameters are and what the focus is, is it Canada and the trucker protests in contrast to the mainstream? Is it a reflection of post-covid predictions for the globe, if so why use the CPI? Be clearer about the parameters of your analysis and how the findings can be generalised out to other contexts.
In the analysis is it difficult to see the narrative analysis and TA (themes 6 and & are more of an exception). Explain how you did the qualitative analysis, why explore predictive/prescriptive threads? Etc. Make it clear to the reader that the themes reflect this analysis and not the original hypothesis, ground the themes in the newspaper articles to build a more credible narrative based on this analysis. When you present the themes you indicate that there is less support for ‘rationality and science’, yet that is the theme you start with, why? Why not tell the story how the data presents it, with the more robust themes and narratives at the beginning and less focus on those with limited support within the data? Likewise, why do the themes which fall outside your lens not get developed. You indicated you were engaging in top down and bottom up analysis, so why leave the themes voiceless? I suggest to refocus the analysis and themes, signpost them more clearly and link them through to the literature they are drawn from.
Author Response
Thank you to the reviewers for their thoughtful and helpful commentary on the manuscript. As they will see, their central comments were most useful to us in our thoroughgoing revision of the manuscript, and we applied their ideas fully and completely. While we remain responsible for the analyses, we believe that the reviewers have helped us greatly in developing a sharper and more focussed article.
- In response to the reviewer’s request, the entire manuscript has been redrafted to improve coherence and the reviewer’s requests for further evidence have been addressed. The manuscript now adds “9/11 and the War on Terror” as a defining event in the contemporary era.
- Following the reviewer, we have explained the national scope of our article and the Canadian context of our analyses. We believe that this further explanation will enable readers to appreciate the distinctions we draw re. the CPI as an index of journalism and commentary in Canada, but not necessarily from Canadian sources and linking Canada increasingly to a global world
- The manuscript now contains an explanation of predictive v. prescriptive threads, as requested by the reviewer. Also, in general, there is a more elaborated discussion of the qualitative analysis. The reviewer asks a good question regarding the underdevelopment of the unanticipated themes. The manuscript now addresses these issues.
Round 2
Reviewer 2 Report
The authors have improved the manuscript with these edits, but there are still some unresolved issues, mainly around the support for the claims the authors make and the focus of the paper, while improved, is still not clear.
The main issue that is confusing is why the authors spend a a considerable about of time describing and discussing the convoy protests in Canada in the introduction and discussion when (a) the period of the research collection is outside the duration of these protests, (b) none of the themes are at all related to these protests. I understand it seems to be an oppositional voice to your findings, but your findings are wholly unrelated to the convoy protests, so why do they receive the amount of space given to them in this article as they do not seem to be central to the research collected or the findings? Thus the focus of the article is still unclear. The authors need to really consider what they are trying to achieve with this manuscript and focus it accordingly.
While you have addressed some of the weaknesses around support for your arguments, there is still large sections of the article were you make contested claims or opinions without any support for them beyond your own opinion.
For example see lines 217 - 224 in the social equality section. Lines 259 - 269, 273-288, 294-299, 303-314 in the local and communitarian sections. Lines 322-330 in the deeply participative section. The authors need to review these sections and support the claims made, as while they may seem obvious and clear to the authors, not all readers will share their viewpoints.
There are some other more minor issues to consider:
Where you use direct quotes you need to include page numbers see lines 227, 290, 449 and add in the page numbers for these quotes.
Proof read and pick up on typo errors, for example see line 113
In Line 746 you refer to the convoy protests being in the current era, but they ended months ago.
You refer to 'radicalised minorities' and the radicalised throughout the manuscript, what do you mean by this label, it isn't a clearly defined term
Author Response
Please see the attached for the review response.
